# The Thermodynamics and Kinetics of a Nitrogen Reaction in an Electric Arc Furnace Smelting Process

**DOI:** 10.3390/ma16010033

**Published:** 2022-12-21

**Authors:** Fujun Zhang, Jingshe Li, Wei Liu, Aoteng Jiao

**Affiliations:** 1State Key Laboratory of Advanced Metallurgy, University of Science and Technology Beijing, Beijing 100083, China; 2School of Metallurgical and Ecological Engineering, University of Science and Technology Beijing, Beijing 100083, China

**Keywords:** electric arc furnace steelmaking, thermodynamics, dynamics, restrictive step, optimization of denitrification process

## Abstract

The nitrogen content of electric arc furnace (EAF) steel is much higher than that of basic oxygen furnace (BOF) steel, which cannot meet the requirements of high-grade steel. Most denitrification processes only considered a single smelting condition, which leads to poor denitrification effect. In this study, a hot state experiment was conducted to simulate the melting process of EAF steelmaking and to explore the thermodynamic and kinetic constraints of the molten steel nitrogen reaction in the scrap melting, oxygen blowing decarburization, and rapid temperature rise stages. The experimental results showed that the nitrogen reaction in the molten pool during the scrap melting stage was a first-order nitrogen absorption reaction, and the reaction-limiting link was the diffusion of nitrogen atoms in the molten steel. When the carbon content increases to 4.5%, the bath temperature decreases to 1550 °C, and the nitrogen partial pressure decreases to 0.2 *P*^Θ^, the nitrogen saturation solubility decreased to 0.0198%, 0.0318%, and 0.0178%, respectively. At the same time, the rate constants decreased to 0.132 m/min, 0.127 m/min, and 0.141 m/min, respectively. The nitrogen reaction in the oxygen blowing decarburization stage was a secondary denitrification reaction, and the reaction-limiting link was the gas–liquid interface chemical reaction. Argon had better degassing effect. When the argon flow rate increased from 100 mL/min to 300 mL/min, the reaction constant increased by about four times. When the oxygen content of molten steel was 0.0260%, the denitrification rate constant decreased by about 2.5 times. The nitrogen content of liquid steel was higher than 0.045%, and the reaction was a secondary reaction. As the nitrogen content decreased, the reaction rate decreased, and the reaction-limiting link changed from the gas–liquid interface chemical reaction to the joint control of mass transfer and chemical reaction. The oxygen content in the molten steel can not only hinder the chemical reaction of nitrogen at the gas–liquid interface, but also reduce the mass transfer rate of nitrogen atoms in the molten steel. The results provided a theoretical basis for the optimization of nitrogen removal process and further reduction of nitrogen content in liquid steel.

## 1. Introduction

Electric arc furnace (EAF) steelmaking is one of the two major steelmaking processes in the world, which is characterized by a short process, low energy consumption, and low carbon emission [1,2,3]. However, because of arc heating, the nitrogen content of EAF steel is (70 × 10^−6^ to 120 × 10^−6^) much higher than that of basic oxygen furnace (BOF) steel (30 × 10^−6^ to 40 × 10^−6^) and cannot meet the performance requirements of high-grade steel, which restricts the development of EAF steelmaking [4].

The final nitrogen content in molten steel is determined by the nitrogen content in the charge and amount of nitrogen removed and absorbed during smelting. As the main raw material of the EAF is scrap steel, the initial nitrogen content is high, and the carbon content is low; therefore, the amount of nitrogen removed during the smelting process is low. In addition, N_2_ molecules in the air of the EAF area are ionized, simplifying the absorption of nitrogen by the EAF steelmaking process. The nitrogen reaction process is a typical gas–liquid interface reaction process, which conforms to Schwartz’s law [5]. N_2_ gas molecules transfer mass through the gas-liquid interface and generate the nitrogen reaction 2 [N] = N_2_. According to the above nitrogen reaction theory, researchers have reported their studies on controlling the nitrogen content of steel during EAF smelting. Harashima [6] studied the effect of blowing argon under reduced pressure and mixed reducing gas or adding iron ore powder at the top of the process on the nitrogen removal rate in molten steel. Neuschütz [7] used methane as the ionized gas and observed that the methane containing arc has the potential to accelerate the removal of dissolved nitrogen. Pal [8] added a directly reduced iron powder to a molten bath and removed the dissolved nitrogen using fine CO bubbles generated deep in the molten bath, which can significantly reduce the nitrogen absorption of molten steel during electrode heating. However, owing to technical problems, these methods are rarely used. In addition, some researchers have studied the influence of different gas injections on the EAF smelting process. Yi [9] and Lv [10] observed that mixing CO_2_ gas in pure oxygen can reduce the “fire point” temperature and iron loss. Wang et al. [11] calculated the detailed process parameters of CO_2_ injection using material and heat balances. Kawakami [12] showed that the nitrogen absorption rate of molten steel injected with nitrogen is controlled by mass transfer in the metal, and the nitrogen absorption rate is controlled by mass transfer in the gas phase when an Ar-N_2_ gas mixture is injected. He [13] and Li [14] observed that CO_2_ injection can increase the stirring effect of the molten pool, which is conducive to nitrogen removal from molten steel. Zhu Rong et al. [15] focused on the influence of bottom-blowing gas type on the nitrogen content of molten steel during EAF steelmaking. When Ar or CO_2_ is used as the bottom-blowing gas in EAF steelmaking, nitrogen can be removed using bubbles, and the nitrogen content of the molten steel decreases. The reaction rate constant of CO_2_ bubble nitrogen absorption is 9.6 times higher than that of Ar, and the nitrogen content of molten steel at the end point of full scrap smelting is ≤0.0045%. In addition, in the actual production process, more pig iron or molten iron was added to increase the carbon content and produce more CO gas, so that the end-point nitrogen content is less than 0.0060%. However, this process increases the carbon emission of EAF smelting process [16,17].

The above research did not consider the influence of the change in the composition, nitrogen partial pressure, and other conditions on the nitrogen reaction during the electric arc furnace smelting process. Most denitrogenating processes only considered a single smelting condition and did not provide specific implementation conditions for denitrogenating process, which reduces the denitrogenating effect and increases the implementation cost. In contrast to previous studies, this work simulated the process conditions at different stages of the electric arc furnace smelting process in a tubular furnace to study the thermodynamic and dynamic characteristics and restrictive links of the nitrogen absorption/denitrification reaction of molten steel to provide a theoretical basis for the best implementation conditions for different denitrogenating processes and reducing the end-point nitrogen content of EAF smelting.

## 2. Materials and Methods

The main iron base materials used in the experiment were 45 steel and industrial pure iron, and their compositions are listed in Table 1. The carbon material was chemically pure graphite powder produced by Sinopharm (Shanghai, China) with a carbon content ≥99.85 wt.%. Ferrous oxide was then added to the steel.

The experimental setup is shown in Figure 1. A Si-Mo heating body was used in the tubular furnace. The maximum temperature of the thermostatic belt (crucible melt area) could reach 1650 °C, and the temperature fluctuation range was ±2 °C, which did not affect the accuracy of the experimental results. To control the bath atmosphere and nitrogen partial pressure during the smelting process, a gas-mixing tank was added. Before the gas was introduced into the molten steel, the gas was mixed in a gas mixing tank with different CO_2_/Ar/N_2_ ratios to regulate the atmosphere or nitrogen partial pressure during the smelting process.

According to the different thermodynamic and dynamic conditions of the EAF smelting process, the entire smelting stage is divided into three stages: scrap melting (molten steel is in the deoxidized state and oxygen content is less than 0.002%), oxygen blowing decarburization (carbon oxygen reaction degassing), and temperature rise (oxygen content ≥ 0.01%). In the scrap melting stage, the molten pool does not supply oxygen, or the oxygen supply intensity is low, and elements such as C, Si, and Mn in the molten steel are not completely oxidized. With an increase in the oxygen supply intensity, the decarburization reaction gradually increases, the furnace is gradually filled with a large amount of flue gas, and the partial pressure of nitrogen in the molten pool decreases. According to the literature [18], when the contents of Si and Mn in molten steel are low, there is no obvious effect on the nitrogen reaction. To supplement the heat and improve the dynamic conditions in the bath, approximately 30% pig iron or molten iron is usually added in the EAF smelting process to increase the carbon content. Therefore, in the scrap melting stage, the influence of carbon content, temperature, and nitrogen partial pressure in the molten steel on the nitrogen reaction was studied. The specific schemes are listed in Table 2. During oxygen-blowing decarburization, the oxygen content in the molten steel increases, the carbon and oxygen reaction is intense, and many CO bubbles are generated in the molten pool. Simultaneously, a large amount of ferrous oxide is generated to promote the formation of slag, cover the surface of the molten pool, and isolate the nitrogen atmosphere. Therefore, in the stage of oxygen blowing decarburization, the influence of the decarburization amount and gas type on the denitrification reaction of molten steel was studied. As the capacity of the tubular furnace was low, an oxygen blowing operation was not conducted. Therefore, the decarburization reaction intensity was simulated by adjusting the flow of argon. The specific schemes are listed in Table 3. During the temperature rise stage, the alloy elements in the molten pool are completely oxidized, oxygen content in the molten steel is high, carbon-oxygen reaction is completed, and stirring conditions of the molten pool are poor. Therefore, the effect of the oxygen content and gas type in the liquid steel on the nitrogen reaction was studied during the rapid heating stage. The experimental scheme is summarized in Table 4. The specific experimental steps are as follows.

(1)High-temperature thermocouples were used to correct the constant-temperature zone area, and the thickness of the bottom plate of the lower furnace of the crucible was adjusted to ensure that the thermocouple in the furnace could accurately display the temperature of the constant-temperature zone.(2)The composition of the molten steel was prepared according to the experimental scheme, and the MgO crucible was placed in the constant-temperature zone. Protective argon was introduced separately, and the flow was set to 600 mL/min for 30 min to purge air in the furnace tube.(3)The heating system was started. After the experimental temperature was reached, the atmosphere or Ar/N_2_ ratio was adjusted according to the experimental scheme, the partial pressure of N_2_ in the furnace was controlled, and nitrogen absorption/denitrification experiments were conducted.(4)A quartz tube was used to absorb the molten steel every 5 min and conduct water-cooling. Hydrogen, oxygen, and nitrogen gas analyzers were used to measure the nitrogen contents of the samples.

## 3. Results and Discussion

### 3.1. Thermodynamic and Kinetic Analyses of the Nitrogen Reaction in the Scrap Melting Stage

#### 3.1.1. Thermodynamic Model of the Nitrogen Absorption Reaction during Scrap Melting

The mode and period of nitrogen absorption and denitrification were the same in the smelting process, and the molten bath had different thermodynamic and kinetic conditions in different periods; therefore, these constraints were manifested as either nitrogen absorption or denitrification. Scrap melting was the main task in the early stages of EAF smelting. The oxygen content in the molten steel was low, and the main impurity elements were not oxidized. The nitrogen partial pressure above the molten pool gradually decreased because of the C–O reaction in the molten pool during oxygen blowing; however, the nitrogen partial pressure in the upper part of the molten pool was high, and the nitrogen reaction at this stage was nitrogen absorption. As shown in Figure 2, Figure 3 and Figure 4, the saturated solubility of nitrogen in the molten steel during scrap melting was affected by the composition, temperature, and partial pressure of nitrogen. The saturated solubility decreased with increasing carbon content, carbon content increased from 0.045% to 2.2%, and saturated solubility decreased from 0.0320% to 0.0198%. A higher temperature was associated with a higher saturated solubility. As the temperature increased from 1550 °C to 1620 °C, the saturated solubility increased from 0.0318% to 0.0332%. With an increase in the nitrogen partial pressure, the nitrogen content in the molten steel gradually increased. In air (*P* = 0.8 *P*^Θ^), the nitrogen saturation solubility in the molten steel was 0.0320%, nitrogen partial pressure decreased to 0.2 *P*^Θ^, and nitrogen saturation solubility in the molten steel was only 0.0178%.

Nitrogen exists in liquid monoatomic form in the liquid metal. The dissolution reaction and equilibrium constant are shown in Equations (1) and (2) [19]. Equation (3) is obtained by taking the logarithm of both sides of the equation. The solubility of gas in steel depends on the gas equilibrium partial pressure, reaction constant, and reaction activity coefficient. The equilibrium constant *K_N_* depends only on the reaction temperature. The *K_N_* value determined by Fujio [20], *lgK* = −518/T − 1.063, was used. *f_N_* is affected by the composition of the molten steel. The activity coefficient was calculated according to Wagner, and the relative activity coefficient is listed in Table 5 to calculate the nitrogen solubility of molten steel under the smelting composition. As shown in Figure 2, Figure 3 and Figure 4, the calculated saturated solubility of nitrogen was consistent with the solubility measured in the experiment, and the experimental value was slightly lower than the calculated value because the reaction struggled to reach the full equilibrium state.
(1)$$\frac{1}{2}N_{2}=\left[N\right]$$
(2)$$K_{N}=\frac{\left[\%N\right]f_{N}}{{\left(P_{N}/P^{\Theta }\right)}^{\frac{1}{2}}}$$
(3)$$lg\left[\%N\right]=1/2lg\left(P_{N_{2}}/P^{\Theta }\right)+lgK_{N}-lgf_{N}$$
where [%*N*] is the nitrogen content in the molten steel, *f_N_* is the nitrogen activity coefficient in the molten steel, *P_N_* is the partial pressure of nitrogen (Pa), and *P*^Θ^ is the standard atmospheric pressure.

#### 3.1.2. Kinetic Model of the Nitrogen Absorption Reaction during Scrap Melting

The nitrogen absorption/denitrification reaction process of molten steel is composed of three steps [15,21,22]:
Gas phase film mass transfer of nitrogen at the interface: $N_{2}\to N_{2}^{*}$. The expression for the correlation reaction rate *V*_1_ is shown in Equation (4).
(4)$$V_{1}=\frac{dw\left[N\right]}{dt}=\frac{k_{N,g}}{RT}\cdot \frac{A}{V_{m}}\cdot \left(P_{N_{2}}-P_{N_{2}}^{*}\right)$$Chemical reaction at the gas–liquid interface: $N_{2}^{*}⇆2N^{*}$. The expression for the related reaction rate *v*_2_ is shown in Equation (5).
(5)$$V_{2}=\frac{dw\left[N\right]}{dt}=\frac{A}{V_{m}}\cdot \left(K_{+}P_{N_{2}}^{*}-K_{-}w{\left[N\right]}^{*2}\right)$$Diffusion of nitrogen atoms in the molten steel: ${\left[N\right]}^{*}⇆\left[N\right]$. The expression for the relevant reaction rate *V*_3_ is shown in Equation (6).
(6)$$V_{3}=\frac{dw\left[N\right]}{dt}=k_{N,l}\cdot \frac{A}{V_{m}}\cdot \left(w{\left[N\right]}^{*}-w\left[N\right]\right)$$
where *k_N,g_* and *k_N,l_* are the mass transfer coefficients of nitrogen in the liquid and gas phases, respectively (m/s); $P_{N_{2}}^{*}$ and $P_{N_{2}}$ are the partial pressures of the liquid phase and interfacial nitrogen, respectively (Pa); *K*_+_ and *K*_−_ are the reaction rate constants of nitrogen absorption and denitrification, respectively; w[*N*] and w[*N*]* are the nitrogen contents of the molten steel and interface, respectively; A is the reaction interfacial area (m^2^); and *V_m_* is the volume of molten steel (m^3^).

The total reaction rate of nitrogen in the molten steel is related to the rate or resistance of the three steps. The slowest rate or largest resistance, which is the total reaction rate, is the limiting step of the nitrogen reaction.

Figure 5 shows the change in the nitrogen content as a function of time under different carbon contents. The nitrogen content in the molten steel increased rapidly within 20 min before nitrogen was introduced. After 25 min, the four carbon contents of the molten steel were saturated, and the nitrogen absorption rate of the molten steel decreased with increasing carbon content. Figure 6 shows the change in the nitrogen content as a function of time under different temperature conditions. The saturated nitrogen content of the molten steel increased as a function of increasing temperature; however, the nitrogen absorption rate of the molten steel at 1620 °C was less than that at 1570 °C during the rapid increase in nitrogen content. As shown in Figure 7, the nitrogen content of the molten steel varied as a function of time under different nitrogen partial pressures. With an increase in the nitrogen partial pressure, the nitrogen saturation solubility in the molten steel increased, and the nitrogen absorption rate increased.

Equation (6) is integrated to obtain Equation (7),
(7)$$ln\frac{w{\left[N\right]}_{e}-w{\left[N\right]}_{0}}{w{\left[N\right]}_{e}-w\left[N\right]}=k_{N,l}\cdot \frac{A}{V_{m}}\cdot t$$
where *w*[*N*]_0_ and *w*[*N*]*_e_* are the initial and saturated nitrogen contents in the molten steel, respectively.

According to the saturated nitrogen content calculated using thermodynamics, the relationship between $ln\frac{w{\left[N\right]}_{e}-W{\left[N\right]}_{0}}{w{\left[N\right]}_{e}-w{\left[N\right]}_{0}}$ and the reaction rate constant *k_N,l_* was plotted in combination with Equation (7). As shown in Figure 8, Figure 9 and Figure 10, the reaction rate constant was directly proportional to $ln\frac{w{\left[N\right]}_{e}-W{\left[N\right]}_{0}}{w{\left[N\right]}_{e}-w{\left[N\right]}_{0}}$, that is, the nitrogen absorption reaction was a first-order reaction, and the diffusion of the liquid boundary layer was the limiting step of the nitrogen absorption reaction. Carbon in the molten steel can increase the saturated solubility of nitrogen in the molten steel but reduce its reaction rate constant. A higher nitrogen saturation solubility was associated with higher carbon content and bath temperature; however, a higher temperature was associated with a lower nitrogen absorption reaction rate constant. The increase in the nitrogen partial pressure only increased the solubility of saturated nitrogen in the molten steel and did not affect the rate constant of the nitrogen absorption reaction.

### 3.2. Thermodynamic and Kinetic Analyses of the Nitrogen Reaction in the Oxygen Blowing Decarbonization Stage

#### 3.2.1. Thermodynamic Analysis of the Nitrogen Removal Reaction in the Oxygen Blowing Decarbonization Stage

In the oxygen blowing decarburization stage, a severe C–O reaction occurred in the molten pool, a large amount of flue gas was above the molten pool, and the nitrogen partial pressure was extremely low. As smelting proceeded, slag formed in the upper part of the molten steel. As shown in Figure 11, when the steel surface was covered with slag, the slag could effectively isolate nitrogen, and the nitrogen content in the molten steel remained unchanged over time. Small vacuum bubbles were formed by the C–O reaction and bottom-blowing gas in the molten steel, and the nitrogen partial pressure at the interface between the molten steel and bubbles was almost zero. Therefore, the nitrogen reaction in the molten pool during the oxygen-blowing decarburization stage is a denitrification reaction.

#### 3.2.2. Kinetics of Denitrification in the Oxygen Blowing Decarburization Stage

As shown in Figure 12, the nitrogen content in the molten steel with different gas flow rates changed as a function of time. With the increase in the gas flow rate, the nitrogen removal rate in the molten steel increased, and the nitrogen removal rate was significantly lower than the nitrogen absorption rate. Figure 13 shows the nitrogen contents of different gases as a function of time under the condition of no oxygen. When the gases were Ar, CO, and CO_2_, the denitrification rate of molten steel decreased in that order. The main reason for this decrease is that the carbon content in the molten steel was low, thus CO_2_ entering the molten steel did not react with carbon in the molten pool; in addition, CO and CO_2_ were dissolved in the molten steel, reducing the content of gas blown into the molten steel in a disguised way, thus the decarburization rate of Ar was the largest.

Equation (8) is obtained by combining Equations (4)–(6) and integral transformation,
(8)$$\frac{1}{w\left[N\right]}-\frac{1}{w{\left[N\right]}_{0}}=K_{-}\cdot \frac{A}{V_{m}}\cdot t$$

According to the calculation results of Equation (8), the relationship between $\frac{1}{w\left[N\right]}-\frac{1}{w{\left[N\right]}_{0}}$ and the reaction rate constant *K_−_* was plotted. As shown in Figure 14 and Figure 15, the denitrogenating reaction rate constant was directly proportional to $\frac{1}{w\left[N\right]}-\frac{1}{w{\left[N\right]}_{0}}$ when the gas was argon, which indicates that the decarburization reaction is a secondary reaction, the interfacial chemical reaction is the limiting step of the denitrogenating reaction, and increasing the gas flow can significantly increase the reaction rate constant. When CO or CO_2_ was injected, the relationship between $\frac{1}{w\left[N\right]}-\frac{1}{w{\left[N\right]}_{0}}$ and the reaction rate constant was directly proportional, and the reaction rate was much lower than that of the argon injection case. Previous studies show that [23,24,25,26] if the carbon content is in the range of 0.0115–0.1942 wt.%, CO_2_ entering the molten steel reacts with carbon to generate CO or decomposes into [C] and [O] in the molten steel, and the oxygen content is maintained in the range of 10^−1^ wt.%. When the carbon content is lower than 0.0115 wt.%, CO_2_ transfers oxygen atoms to steel, and the mass fraction of oxygen is 0.1977 wt.% in the stable state. After CO enters the molten steel, the carbon oxygen concentration is less than the carbon oxygen solubility product under the condition of gas-phase equilibrium, thus CO gas will dissolve in the molten steel and release carbon and oxygen elements. This dissolution results in the volume of gas in the molten steel during CO gas injection being smaller than the Ar volume at the same bottom-blowing flow, and the CO denitrification capacity was slightly smaller than that of Ar.

### 3.3. Thermodynamic and Kinetic Analyses of the Nitrogen Reaction in the Rapid Heating Stage

#### 3.3.1. Thermodynamic Analysis of the Nitrogen Reaction in the Heating Stage

Impurity elements in molten steel are completely oxidized by the oxygen blowing smelting process, and the oxygen content in molten steel increases. The previous shows that [27] oxygen is the surface-active substance in molten steel. When the oxygen content in molten steel is high, this element is concentrated on the surface of molten steel and occupies surface binding sites that prevent nitrogen adsorption at the interface. According to the surface position closure model, the calculation formula for the nitrogen reaction rate constant is given by Equation (9) [28]. The experimental results show that a higher oxygen content was associated with a higher blocking effect. When the oxygen content was greater than 0.0300%, the molten steel could barely absorb nitrogen.
(9)$$k=k^{*}\left(1-\theta \right)$$
where *k** is the rate constant of the pure iron liquid and *θ* is the fraction of the surface occupied by oxygen, which was calculated according to the Langmuir isotherm and oxygen activity.

#### 3.3.2. Kinetic Analysis of the Nitrogen Reaction in the Heating Stage

Figure 16 shows the change in the nitrogen content as a function of time under different oxygen contents in the molten steel. With an increase in the oxygen content in the molten steel, the denitrification rate decreased, and an increase in the blowing time resulted in a slow decrease in the denitrification rate. Figure 17 shows the change in the nitrogen content in the molten steel under different gas conditions with an oxygen content of 0.0260%. When CO_2_ gas is injected into molten steel with a high oxygen content, the nitrogen content in the molten steel remains unchanged. Previous studies show that [23,24,25,26] when the oxygen content in the molten steel is high and carbon content is low, CO_2_ gas blown into the molten steel is completely dissolved, leading to an “oxygen and carbon increase” phenomenon, and vacuum bubbles cannot be formed in the molten steel. When the injection gas was CO, the higher oxygen content in the steel inhibited the dissolution of CO, and the gap between the denitrification capacities of CO and Ar was narrower than that of CO denitrification under oxygen-free conditions.

The relationship between $\frac{1}{w\left[N\right]}-\frac{1}{w{\left[N\right]}_{0}}$ and the reaction rate constant is described by Equation (8). As shown in Figure 18 and Figure 19, the denitrification rate decreased after an increase in the oxygen content of the molten steel. $\frac{1}{w\left[N\right]}-\frac{1}{w{\left[N\right]}_{0}}$ is directly proportional to the reaction rate constant; that is, the denitrification reaction is a secondary reaction, and the interfacial chemical reaction is the denitrification-limiting step. With an increase in the oxygen content of the molten steel, the reaction constant decreased. When the oxygen content in the molten steel was high, oxygen was enriched at the interface, occupying nitrogen binding sites and reducing the denitrification reaction rate; the reaction constant was lower than that of the denitrification reaction rate constant under oxygen-free conditions. At a high oxygen content, the reaction rate constant was directly proportional to time 45 min before Ar injection; however, the reaction rate constant started to decrease after 45 min, and the reaction limiting steps are liquid phase mass transfer and interfacial reaction. The turning point of the CO reaction rate constant for the gas injection occurred 30 min in advance. As indicated by this analysis, the oxygen element in molten steel not only occupies the nitrogen binding sites on the interface and limits the interfacial reaction, but also a high oxygen content reduces the mass transfer rate of nitrogen atoms in the molten steel.

### 3.4. Optimization of the Nitrogen Removal Process in EAF Smelting

During oxygen blowing and electrode heating in the EAF smelting process, two special thermodynamic and kinetic regions are formed in the molten pool. The oxygen jet impinges on the liquid steel surface and discharges the molten slag on the surface of the liquid steel to form pits, commonly known as “fire spots”. A violent oxidation reaction occurs in the fire point area, producing a large amount of CO gas at temperatures up to 2400–2600 °C. A previous study shows that [27] when the temperature reaches 2130 °C, the resistance of the oxygen content in molten steel to nitrogen at the interface disappears. Therefore, the nitrogen reaction at the fire point is a nitrogen removal reaction.

The nitrogen dissolution process is microscopically divided into two steps [28]:The nitrogen molecule *N*_2_ absorbs energy and becomes an activated molecule of an isomeric form, as shown in Equation (10).
(10)$$\frac{1}{2}N_{2}+E_{a}=N^{*}$$Active molecules dissolve into molten steel.
(11)$$N^{*}=N$$

The arc generated by electrode heating is a gas discharge phenomenon, which releases a large amount of energy, and the center temperature of the arc is approximately 10,000 °C higher. In the arc area, nitrogen molecules in the air are excited and ionized by the high-temperature arc to generate large amounts of *N*_2_, *N*, *N*^+^, $N_{2}^{+}$, *N**, *N*^−^, and high-energy particles. Because its activity and energy are improved, the arc-heating zone is characterized by rapid nitrogen absorption.

In summary, in the early stage of the EAF smelting process, the main task is to melt the scrap, and the carbon–oxygen reaction in the molten pool is weak, nitrogen partial pressure in the upper part of the molten steel is high, nitrogen content in the molten pool is low, equilibrium nitrogen partial pressure is small, and driving force of nitrogen absorption is large. The nitrogen reaction is a nitrogen absorption reaction. In addition, the nitrogen molecules in the excited and ionized air in the arc heating zone contain large amounts of *N*_2_, *N*, *N*^+^, $N_{2}^{+}$, *N**, *N*^−^, and high-energy particles that increase the reaction rate of nitrogen absorption, thus the nitrogen content in the molten steel during scrap melting almost reaches the saturation state. The above research shows that the nitrogen absorption reaction is a first-order reaction, and the mass transfer of nitrogen atoms at the gas–liquid interface is a reaction-limiting step. After slagging, the C–O reaction of the molten steel gradually intensified, nitrogen partial pressure at the interface of the molten steel decreased, and many vacuum bubbles were generated in the molten steel. The nitrogen reaction is a denitrification reaction, which is a secondary reaction, and the chemical reaction at the gas–liquid interface is a limiting step. Scrap was used as the raw material for EAF smelting, but its carbon content was low. Compared with the converter, there was no carbon–oxygen reaction in the molten bath during the EAF smelting process, and the amount of small vacuum bubbles generated was small. CO and CO_2_ were used as bottom-blowing gases, which dissolved into the molten steel, increasing the oxygen and carbon contents in the steel. The nitrogen removal capacity was slightly lower than that of Ar gas. During the heating stage, the oxygen content in the molten steel was high, which not only inhibited the gas–liquid interface reaction rate, but also reduced the mass transfer rate of nitrogen atoms in the molten steel. The limiting step of the denitrification reaction changed to joint control by the nitrogen interfacial mass transfer and chemical reaction. The scrap melting period is a nitrogen absorption process during the entire smelting process, and the nitrogen absorption rate and amount were much higher than the subsequent nitrogen removal amount. Therefore, during scrap smelting, maintaining a higher carbon content in molten steel or a lower molten pool temperature is used to reduce the saturated solubility and reaction rate constant of nitrogen in molten steel. Pre-slag making, slag retention, and continuous feeding should be used to isolate the nitrogen atmosphere at the molten steel interface, which can effectively reduce the scrap melting period and amount of nitrogen absorption in the molten steel. As oxygen is the active substance in the molten steel, the nitrogen reaction rate at the gas–liquid interface and mass transfer rate in the molten steel decreased, thus the nitrogen removal efficiency was low. The oxygen content of the molten steel was lower during the oxygen-blowing decarbonization period. The rate constant of the denitrification reaction and mass transfer coefficient of nitrogen in the molten steel can be increased by increasing gas generation.

## 4. Conclusions

The composition of liquid steel and partial pressure of nitrogen changed constantly, and the change trend of nitrogen content in liquid steel was different. According to the different thermodynamic and dynamic conditions of the electric arc furnace smelting process, the whole smelting stage was divided into three stages: scrap melting stage, oxygen blowing decarburization stage, and heating up stage. The limiting links of nitrogen reaction in different smelting stages were different, which provided the best implementation conditions for different denitrogenating processes and further reduced the nitrogen content of electric arc furnace steel. However, this study did not involve the solution of nitrogen absorption in charging process and arc heating process, and the nitrogen content in liquid steel was still difficult to reach the level of converter steel. The specific research results are as follows:(1)During the scrap melting period, the nitrogen partial pressure in the upper part of the molten steel was high, and the upper part of the molten steel was not covered by liquid slag. The gas–liquid interface was in full contact, and the nitrogen molecules in the air were ionized by the arc heating zone to *N*_2_, *N*, *N*^+^, $N_{2}^{+}$, *N**, *N*^−^, and high-energy particles to improve the nitrogen reaction rate. The nitrogen content was close to saturation. The nitrogen absorption reaction is a first-order reaction, and the mass transfer of nitrogen atoms at the gas–liquid interface is a reaction-limiting step. Increasing the carbon content in the molten steel and reducing the temperature of the molten pool can reduce the solubility of nitrogen in the molten steel and reaction rate constant. Alternatively, advanced slagging, using the slag retention process with continuous feeding and isolating the nitrogen atmosphere on the steel surface, can effectively reduce the amount of nitrogen absorption.(2)During the oxygen blowing decarburization period, the nitrogen partial pressure in the upper part of the molten steel was low, and the liquid surface was well-covered with slag, which could effectively isolate the gas. The nitrogen reaction was characterized by denitrification. The denitrification reaction is a secondary reaction, and the chemical reaction at the gas–liquid interface is a limiting step. Controlling the oxygen content in molten steel and increasing gas generation can improve the denitrification reaction rate constant and nitrogen mass transfer coefficient in the molten steel.(3)During the heating stage, the oxygen content in the molten steel was high, which not only inhibited the gas–liquid interfacial reaction rate, but also reduced the mass transfer rate of nitrogen atoms in the molten steel. The limiting step of the denitrification reaction changed to the joint control by the nitrogen interfacial mass transfer and chemical reaction. When CO or CO_2_ was used as the denitrification gas, the gas partially dissolved into the molten steel, and the efficiency of the gas blowing denitrogenating at this stage was low.

## Figures and Tables

**Figure 1 materials-16-00033-f001:**
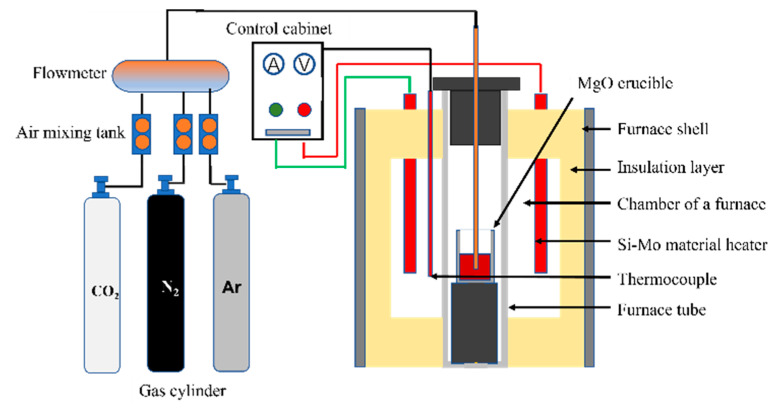
Schematic diagram of the experimental device.

**Figure 2 materials-16-00033-f002:**
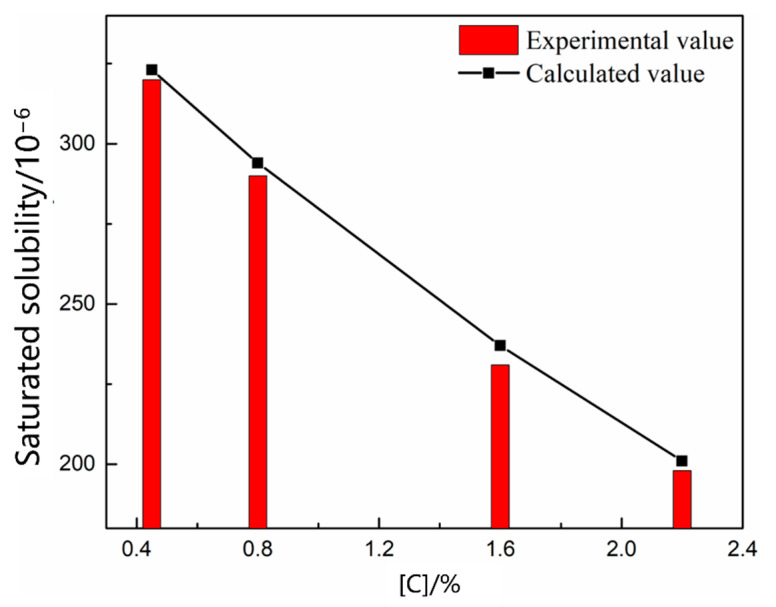
Effect of carbon content on nitrogen solubility.

**Figure 3 materials-16-00033-f003:**
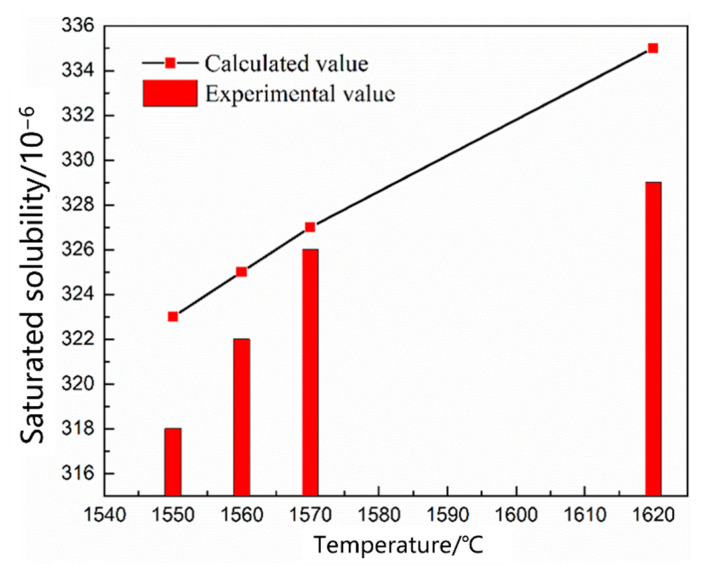
Effect of temperature on nitrogen solubility.

**Figure 4 materials-16-00033-f004:**
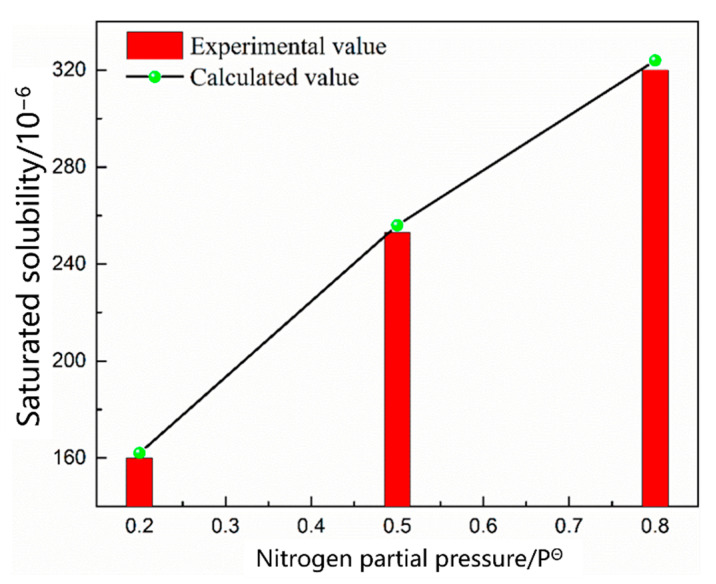
Effect of nitrogen partial pressure on nitrogen solubility.

**Figure 5 materials-16-00033-f005:**
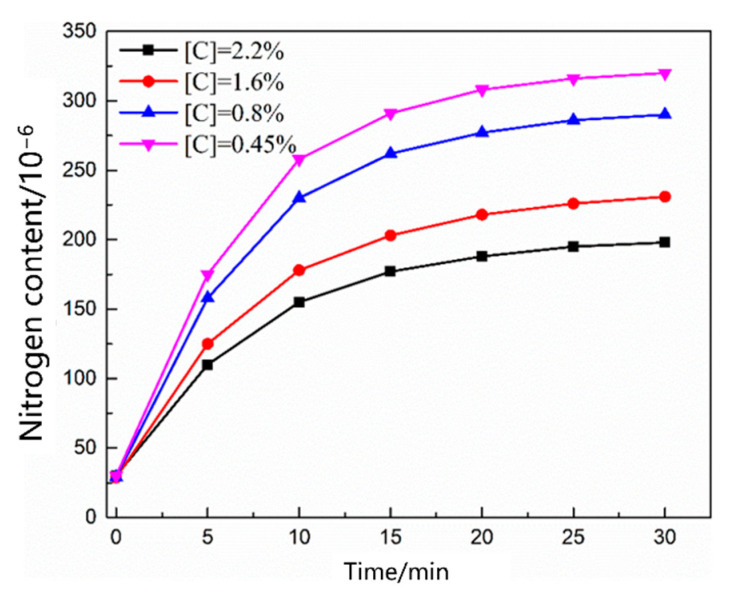
Change in the nitrogen content in the liquid steel with different carbon contents.

**Figure 6 materials-16-00033-f006:**
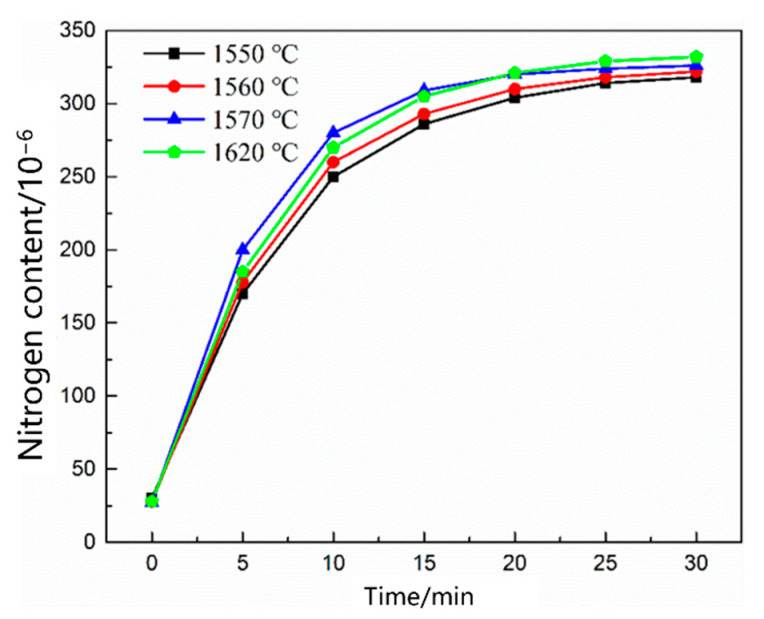
Change in the nitrogen content in the liquid steel at different temperatures.

**Figure 7 materials-16-00033-f007:**
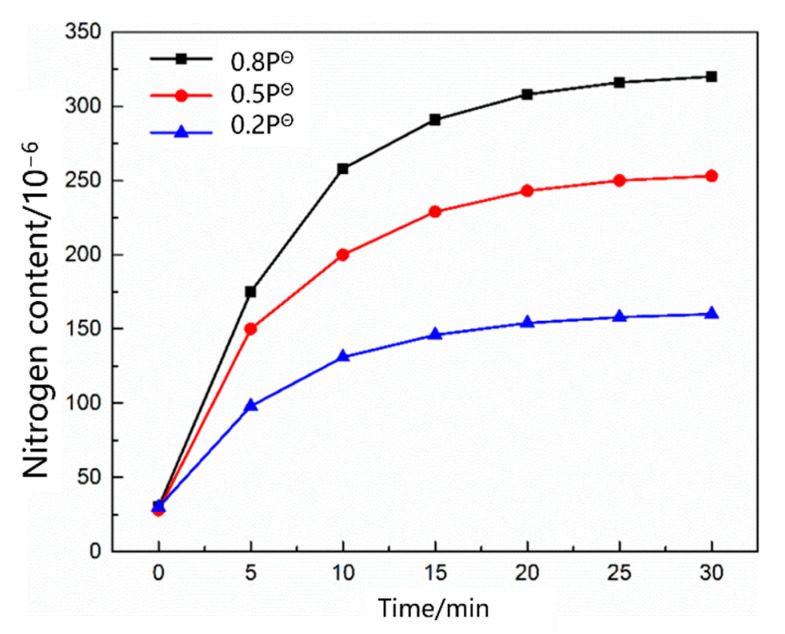
Change in the liquid nitrogen content in steel with different nitrogen partial pressures.

**Figure 8 materials-16-00033-f008:**
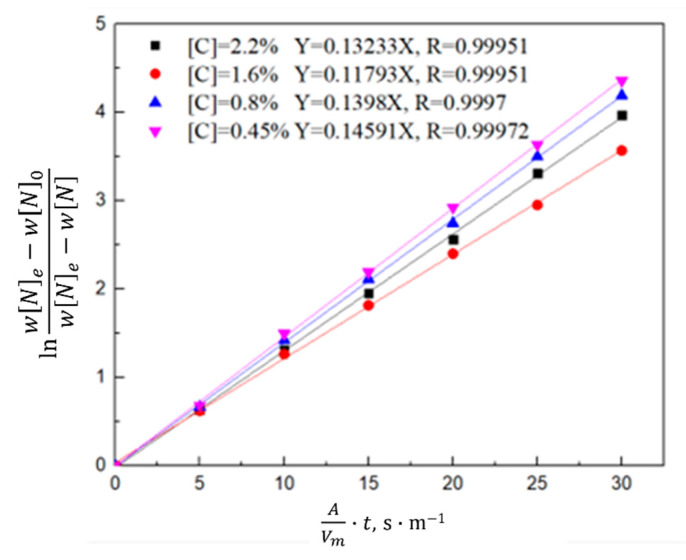
Kinetics of the nitrogen amount in the molten steel with different carbon contents.

**Figure 9 materials-16-00033-f009:**
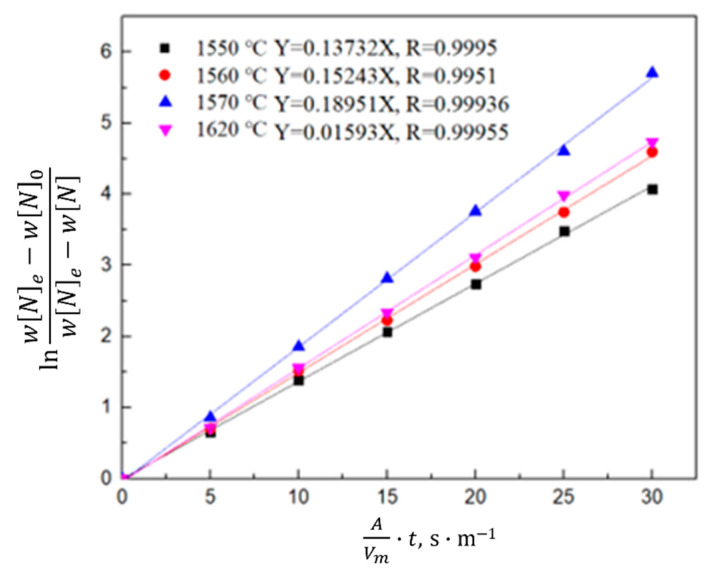
Kinetics of the nitrogen amount in the molten steel at different temperatures.

**Figure 10 materials-16-00033-f010:**
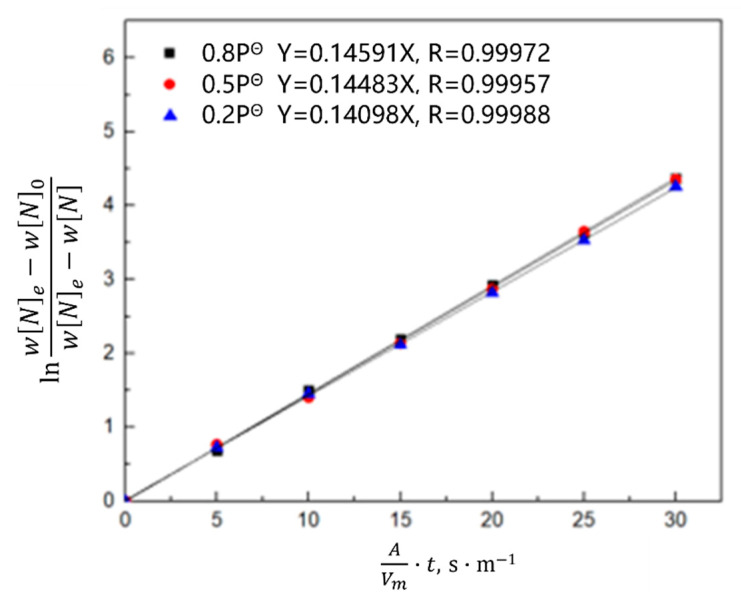
Kinetics of the nitrogen amount in the molten steel under different nitrogen partial pressures.

**Figure 11 materials-16-00033-f011:**
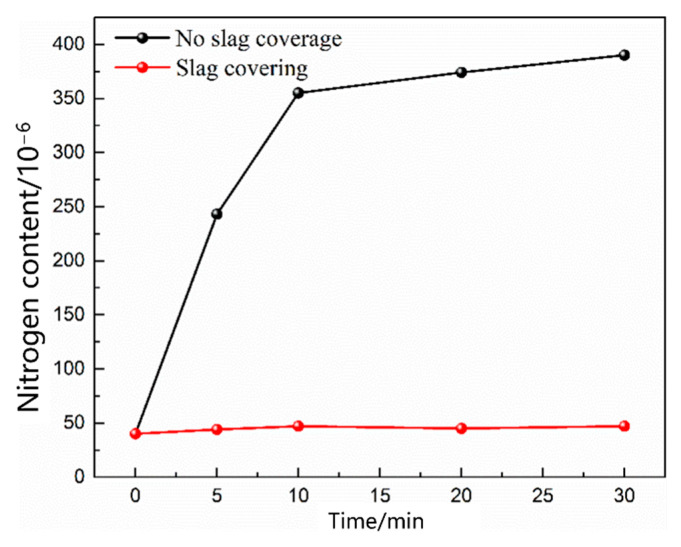
Effect of slag covering on the nitrogen content in the molten steel.

**Figure 12 materials-16-00033-f012:**
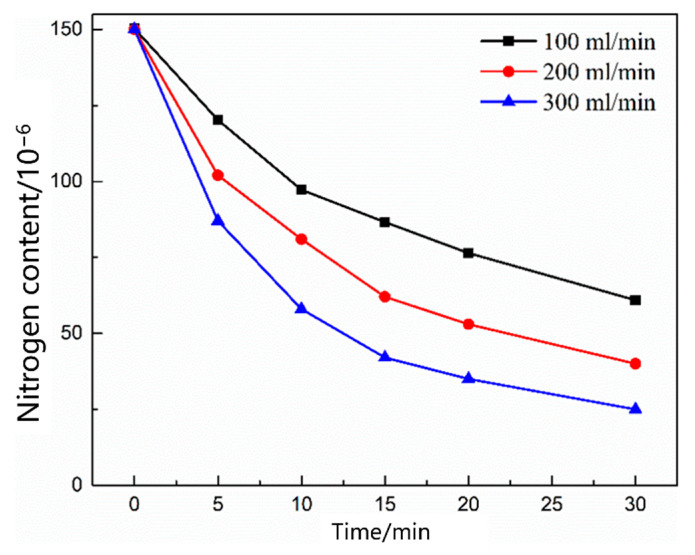
Change in the nitrogen content in the liquid steel under different flow rates.

**Figure 13 materials-16-00033-f013:**
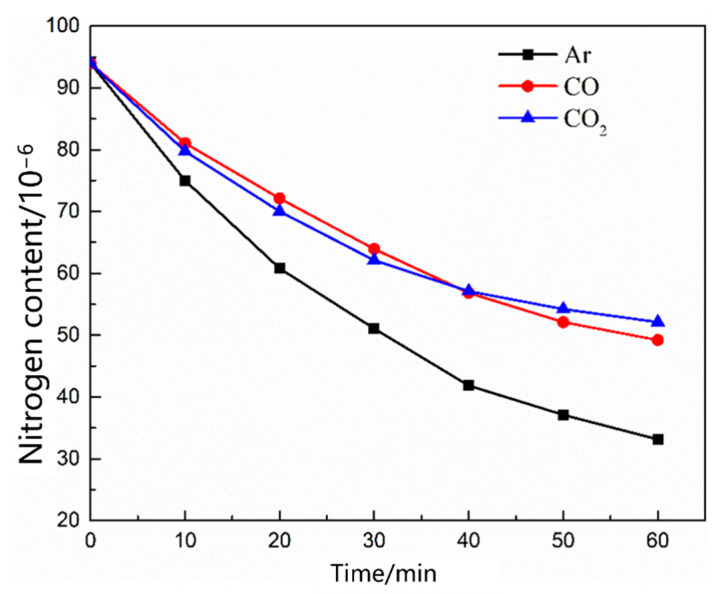
Change in the nitrogen content in the different gases of the oxygen-free molten steel.

**Figure 14 materials-16-00033-f014:**
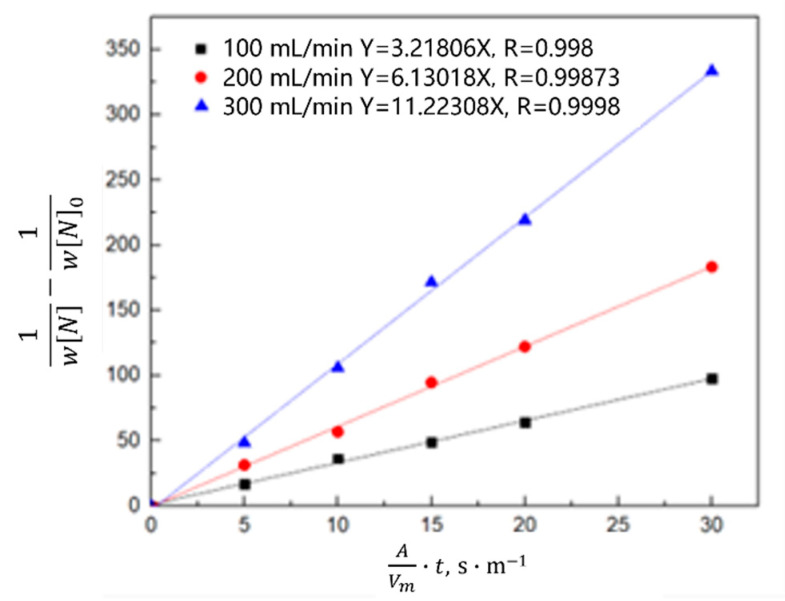
Kinetics of nitrogen removal from the molten steel at different flow rates.

**Figure 15 materials-16-00033-f015:**
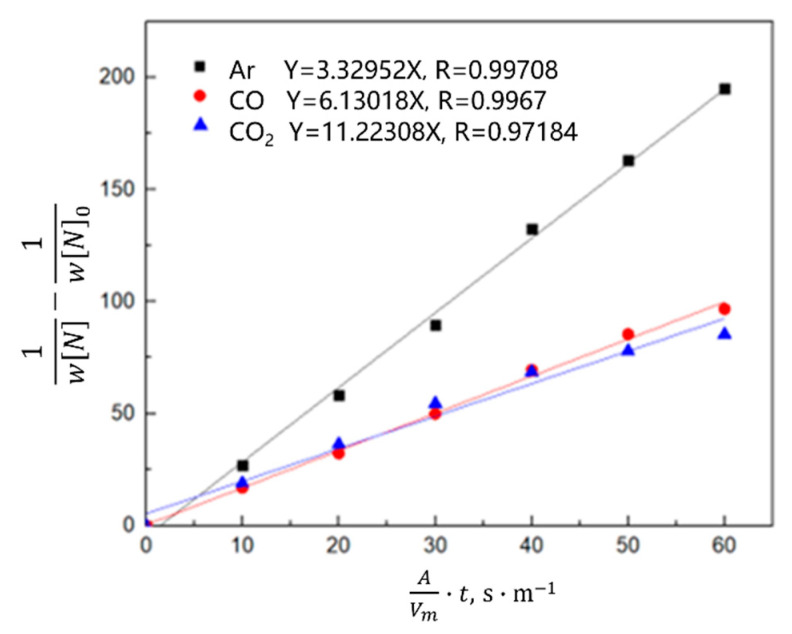
Kinetics of nitrogen removal from the oxygen-free molten steel with different gases.

**Figure 16 materials-16-00033-f016:**
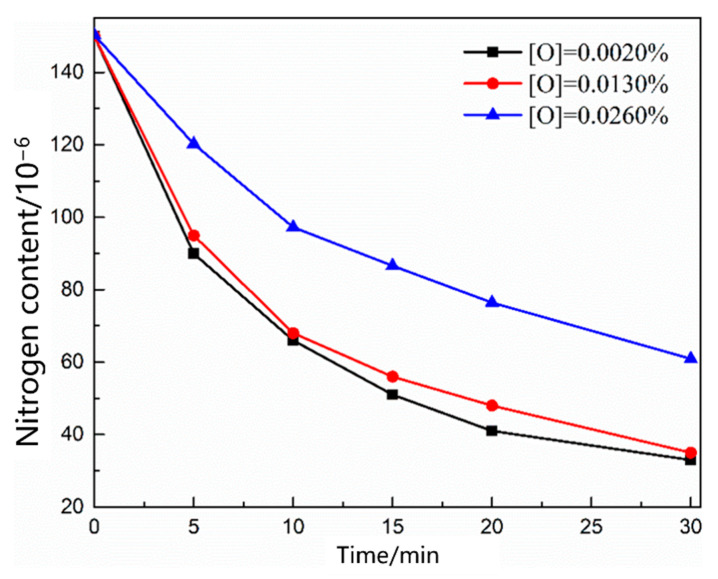
Change in the nitrogen content in the liquid steel with different oxygen contents.

**Figure 17 materials-16-00033-f017:**
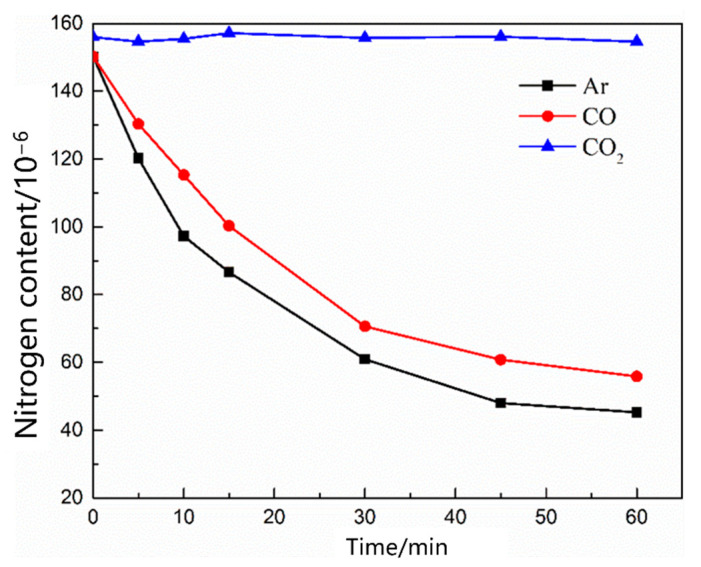
Change in the nitrogen content in the liquid steel with 0.026% oxygen content injected with different gases.

**Figure 18 materials-16-00033-f018:**
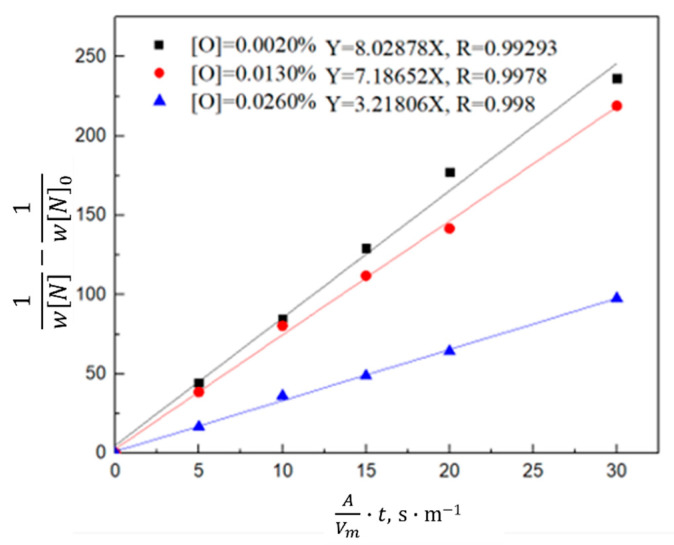
Kinetics of nitrogen removal from the molten steel with different oxygen contents.

**Figure 19 materials-16-00033-f019:**
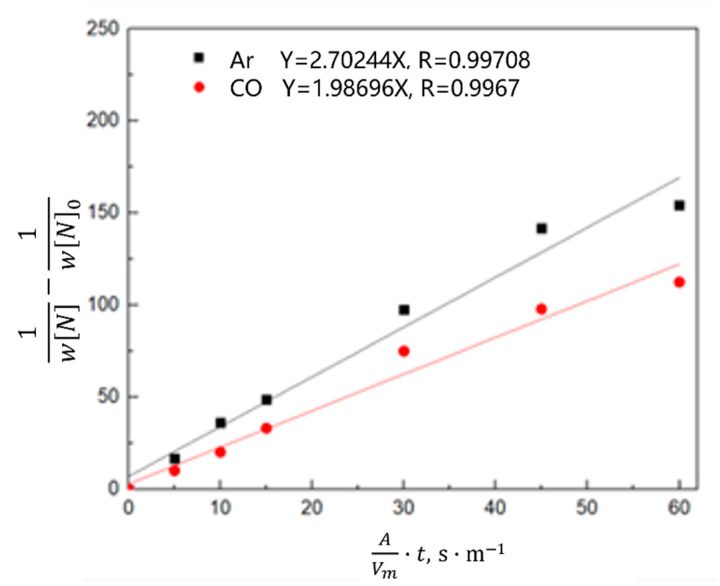
Kinetic law of nitrogen absorption by molten steel with a 0.026% oxygen content and different gases.

**Table 1 materials-16-00033-t001:** Chemical composition of 45 steel.

Element	C	Si	Mn	S	N
Content/wt.%	0.45	0.2	0.75	0.03	0.0045

**Table 2 materials-16-00033-t002:** Experimental plan for scrap melting period.

NO.	Carbon Content/%	Temperature/°C	Nitrogen Partial Pressure/*P*^Θ^
1	0.45	1550	0.8
2	0.8		
3	1.6		
4	2.2		
5	0.45	1550	0.8
6		1560	
7		1570	
8		1620	
9	0.45	1550	0.2
10			0.5
11			0.8

**Table 3 materials-16-00033-t003:** Experimental plan for oxygen blowing decarburization period.

NO.	Gas Flow/mL/min	Gas Type	Slag
1	100	Ar	--
2	200		
3	300		
4	100	Ar	--
5	100	CO	
6		CO_2_	
7		N_2_	Cover

**Table 4 materials-16-00033-t004:** Experimental plan for rapid heating period.

NO.	Gas Type	Oxygen Content/%
1	Ar	0.002
2		0.013
3		0.026
4	Ar	0.026
5	CO	
6	CO_2_	

**Table 5 materials-16-00033-t005:** Relative activity coefficient.

i	N	C	Si	Mn	S	N
$e_{N}^{i}$	0.048	0.118	0.043	−0.024	0.007	0.005

## Data Availability

Not applicable.

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
