# Peer review of "The Thermodynamics and Kinetics of a Nitrogen Reaction in an Electric Arc Furnace Smelting Process"

_materials, 2022, doi:10.3390/ma16010033_

Round 1

Reviewer 1 Report

The authors submitted “Thermodynamics and Kinetics of a Nitrogen Reaction in an Electric Arc Furnace Smelting Process” for publication in “Materials”.

Here is my comments:

1.    The text needs careful revision with respect to grammar and format (spaces, coma, …etc.), the style of the journal must be checked with respect of key words.

2.    The authors should increase the references with number of references and update.

3.    In conclusion, this paper can be accepted with minor revision.

Author Response

Dear editors and reviewers: Thank you for your letter and reviewer's comments on our manuscript entitled "Thermodynamics and kinetics of nitrogen reactions in electric arc furnace smelting". The reviews are valuable and very helpful. We read the comments carefully and made corrections. Following the instructions in your letter, we uploaded the files for the revised manuscript. Replies to reviewer comments are marked in red, as shown below. We very much appreciate your permission to resubmit a revised copy of the manuscript, and we greatly appreciate your time and consideration. sincerely. Zhang Fujun. Reviewer #1: Question 1. The text needs to be carefully revised for grammar and formatting (spaces, commas, ... etc.), and the style of the journal must be checked against key words. Response: The author read the manuscript carefully and revised some grammar and formatting in the text. Question 2. The authors should increase the number of references and update them. Response: The authors have added some references to the manuscript to illustrate the importance and reliability of this study. Thanks again for your valuable comments and reviewers! At the same time, we hope that if you find any deficiencies during the review process, please notify us in time, and hope that the article will be published as soon as possible. Thanks again!

Reviewer 2 Report

The paper is well-written and has a high potential to be accepted. However, the following comments must be considered.

1) The number of papers discussed in the introduction is low, please make more discussions.

2) In the conclusions, please first describe everything and then start the salient remarking points.

Author Response

Dear editors and reviewers: Thank you for your letter and reviewer's comments on our manuscript entitled "Thermodynamics and kinetics of nitrogen reactions in electric arc furnace smelting". The reviews are valuable and very helpful. We read the comments carefully and made corrections. Following the instructions in your letter, we uploaded the files for the revised manuscript. Replies to reviewer comments are marked in red, as shown below. We very much appreciate your permission to resubmit a revised copy of the manuscript, and we greatly appreciate your time and consideration. sincerely. Zhang Fujun. Reviewer #2: Question 1. The number of papers discussed in the introduction is small, please discuss more. Response: In the Introduction, some references are added and discussed in detail. Question 2. In the conclusion, please describe everything first, then start the highlighted comment points. Answer: In the conclusion of the paper, the description of the research content is supplemented.

Reviewer 3 Report

The manuscript reports the Thermodynamics and Kinetics of a Nitrogen Reaction in an Electric Arc Furnace Smelting Process. Electric arc furnace (EAF) steelmaking is characterized by a short process, low energy consumption, and low carbon emissions. The manuscript may accept after major revision.

1.     The main objective of the work is not clearly stated in the abstract. Please make it clear.

2.     The authors should explain the importance of the work in detail in order to attract the readership of this journal.

3.     The authors should highlight the Hypothesis, Experiments, and Findings (some of the numerical values). The preferred format for the Abstract should be used in order to attract the readership of this journal.

4.     Kindly provide more significant data which emphasizes the importance of the work.

5.     More different isotherm models should be included in the revised manuscript such as Frumkin, Freundlich and etc….

6.     Surface morphology by SEM analysis before and after decarburization should be included

7.     Abbreviations should be defined in the first appearance. Please revise.

8.     Conclusion Good but please include the limitations and what can be done for future studies.

9.     Language needs substantial improvement.

10.  All equations in the manuscript need citation.

11.  Some figures could be merged in the revised manuscript i.e. 14 and 15.

Author Response

Dear editors and reviewers:

Thank you for your letter and reviewer's comments on our manuscript entitled "Thermodynamics and kinetics of nitrogen reactions in electric arc furnace smelting". The reviews are valuable and very helpful. We read the comments carefully and made corrections. Following the instructions in your letter, we uploaded the files for the revised manuscript. Replies to reviewer comments are marked in red, as shown below.

We very much appreciate your permission to resubmit a revised copy of the manuscript, and we greatly appreciate your time and consideration.

sincerely.

Zhang Fujun.

Reviewer #3:

Question 1. The main goal of this work is not clearly stated in the abstract. Please make it clear.

Response: The main purpose of this study is explained by re-editing the abstract of the paper.

Question 2. The authors should explain in detail the significance of the work in order to appeal to the journal's readership.

Response: The Introduction has been revised to explain in detail the significance of this study.

Question 3. The authors should highlight hypotheses, experiments, and findings (some numerical values). The preferred format for abstracts should be used in order to appeal to the journal's readership.

A: The abstract of the paper has been re-edited to emphasize the findings of this study with more experimental data.

Question 4. Please provide more significant data emphasizing the importance of the work.

Response: In the Introduction, more data have been added to emphasize the importance of this study.

Question 5. More different isotherm models such as Frumkin, Froundlich, etc. should be included in the revised manuscript.

Response: The influence of the chemical composition of molten steel and nitrogen partial pressure on the saturated solubility of molten steel under constant temperature conditions is mainly studied, and the comparison between different models is not involved. Therefore, in order to highlight the research objectives and avoid too long articles, it is not appropriate to consider more isothermal models.

Question 6. Surface topography by SEM analysis before and after decarburization should be included.

Response: In the scrap melting stage, the effect of the carbon content of molten steel on the saturated solubility of molten steel was studied, without involving the surface morphology of the scrap melting process. More importantly, the experimental materials use steel and graphite with accurate chemical composition, which can ensure the accuracy of the carbon content of each group of experimental materials. There is no need to use SEM to analyze the surface morphology before and after decarburization.

Question 7. Abbreviations should be defined at the first occurrence. Please revise.

Response:The abbreviations appearing for the first time in the text are defined.

Q8. Conclusion Good but please include the limitations and what can be done for future studies.

Response:The conclusion part of the paper is supplemented, and the limitations and application prospects of this study are described.

Q9. Language needs substantial improvement.

Response:The author carefully read the manuscript and revised some grammar and format in the text.

Q10. All equations in the manuscript need citation.

Response:The equations in the paper are supplemented with references.

Q11. Some figures could be merged in the revised manuscript i.e. 14 and 15.

Answer: The figure in the article shows the change trend of nitrogen content in molten steel under univariate conditions, and does not involve multivariate comparison. After merging, it is easy for readers to misunderstand, so similar figures in the text are not combined.

Round 2

Reviewer 3 Report

All comments have been addressed, now the manuscript could be accepted